# Performance of HIV rapid testing algorithm in Nigeria: Findings from a household-based Nigeria HIV/AIDS Indicator and Impact Survey (NAIIS)

Hetal K. Patel[1]*, Sunday Ikpe[2], Megan Bronson[1], Sehin Birhanu[1], Alash'le Abimiku[2], Ibrahim Jahun[1], Mervi Detorio[1], Kathryn Lupoli[1], Daniel Yavo[1], Orji O. Bassey[1], Tapdiyel D. Jelpe[1], Brian Kagurusi[2], Nnaemeka C. Iriemenam[1], Divya Patel[1], McPaul I. Okoye[1], Ibrahim T. Dalhatu[1], Stephen Ohakanu[2], Andrew C. Voetsch[1], Sani Aliyu[3], Gregory Ashefor[3], Aliyu Gambo[3], Gabriel O. Ikwulono[4], Charles Nzelu[4], Isaac F. Adewole[4,5], Mahesh Swaminathan[1], Bharat Parekh[1]*

1 Centers for Disease Control and Prevention, Atlanta, Georgia, United States of America, 2 University of Maryland School of Medicine Institute of Human Virology, Baltimore, Maryland, United States of America, 3 Nigeria AIDS Control Agency, Abuja, Nigeria, 4 Federal Ministry of Health, Abuja, Nigeria, 5 College of Medicine, University of Ibadan, Ibadan, Nigeria

* hpatel@cdc.gov (HKP); bsp1@cdc.gov (BP)

**Data Availability Statement:** All relevant data are within the paper and its Supporting Information files.

## Abstract

### Background

The Nigeria AIDS Indicator and Impact Survey (NAIIS), a cross-sectional household survey, was conducted in 2018 with primary objectives to estimate HIV prevalence, HIV-1 incidence, and status of UNAIDS 90-90-90 cascade. We conducted retrospective analysis of the performance of HIV rapid tests and the national HIV testing algorithm used in Nigeria.

### Methods

The national algorithm included Determine HIV-1/2 as test 1 (T1), Unigold HIV-1/2 as test 2 (T2), and StatPak HIV-1/2 as the tie-breaker test (T3). Individuals reactive with T1 and either T2 or T3 were considered HIV-positive. HIV-positive specimens from the algorithm were further confirmed for the survey using supplemental test Geenius HIV-1/2. If Geenius did not confirm HIV-positive status, HIV-1 Western blot was performed. We calculated the concordance between tests and positive predictive value (PPV) of the algorithm on unweighted data.

### Results

Of 204,930 participants (ages ≥18 months) 5,103 (2.5%) were reactive on T1. Serial testing of T1 reactive specimens with T2 or if needed by tiebreaker T3 identified 2958 (1.44%) persons as HIV-positive. Supplemental testing confirmed 2,800 (95%) as HIV-positive (HIV-1 = 2,767 [98.8%]; HIV-2 = 5 [0.2%]; dual infections = 22 [0.8%]). Concordance between T1 and T2 was 56.6% while PPV of the national algorithm was 94.5%.

**Funding:** This project is supported by the President's Emergency Plan for AIDS Relief (PEPFAR) through the Centers for Disease Control and Prevention (CDC) under the terms of Cooperative Agreement #U2GGH002108 to the University of Maryland, Baltimore and by the Global Fund to Fight AIDS, Tuberculosis, and Malaria through the National Agency for the Control of AIDS, Nigeria, under the contract #NGA-H-NACA to the University of Maryland, Baltimore. The findings and conclusions of this report is those of the authors and do not necessarily represent the official position of the funding agencies. The funders had no role in study design, data collection and analysis, decision to publish, or preparation of the manuscript.

**Competing interests:** The authors have declared that no competing interests exist.

## Conclusions

Our results show high discordant rates and poor PPV of the national algorithm with a false-positive rate of about 5.5% in the NAIIS survey. Considering our findings have major implications for HIV diagnosis in routine HIV testing services, additional evaluation of testing algorithm is warranted in Nigeria.

## Introduction

Globally HIV rapid test (RT)-based testing algorithms are widely used for diagnosis of HIV. Generally, due to the performance characteristics of HIV RT and the added operational benefits, such as ease of use and interpretation, storage at ambient temperature and no specialized laboratory equipment requirements, RTs have become a preferred technique for HIV diagnosis in low- and middle-income countries. By bringing testing out of laboratories and closer to communities, HIV-positive clients can immediately know their status and initiate proper care and treatment. In the United States, the 2016 guidance for non-clinical settings outlined multiple testing options, including rapid HIV testing, with the aim to immediately enroll presumptive reactive individuals to care [1].

HIV RTs used in programs supported by the President's Emergency Plan for AIDS Relief (PEPFAR) must meet high-performance characteristics and are evaluated and approved for use under the US Agency for International Development (USAID) List of Approved HIV/AIDS Rapid Test Kits and the WHO Prequalification of Medicines Programme (PQ) (https://www.usaid.gov/sites/default/files/documents/1864/HIV_RTK_May_1_2020.pdf). Under these processes, the minimum performance requirements of any HIV RT are sensitivity of ≥99% and specificity of ≥98%. For HIV diagnosis, three consecutive reactive tests should be used for a HIV-positive diagnosis and must be reviewed in the context of HIV prevalence because the overall positive predictive value (PPV) is expected to be low in a low prevalence settings [2, 3].

Positive confirmation continues to be even more important as we expand initiation on antiretroviral therapy (ART) immediately after receiving HIV-positive test results [4]. The false positive rate observed by a testing algorithm can be significantly reduced by including a third HIV RT or by including a confirmatory test such as Geenius HIV-1/2 RT (Geenius) or HIV-1 Western blot (WB) [5–8]. However, the Geenius and WB tests are more complex, expensive, and require appropriate laboratory infrastructure and instrument maintenance. The Geenius assay is a rapid cartridge format and has more feasibility of use at the near point of care, although it requires trained laboratory technicians and use of an automated reader for final interpretation.

A national review of HIV testing algorithms for use in Nigeria was conducted in 2015 [9]. A serial testing algorithm comprising of Determine HIV-1/2, Unigold HIV-1/2 and StatPak HIV-1/2 (as a tie breaker test) was selected. We report here an analysis of the national testing algorithm and its performance using Nigeria AIDS Indicator and Impact Survey (NAIIS) 2018 data.

## Methods

Similar to the Population-based HIV Impact Assessment (PHIA) project (https://phia.icap.columbia.edu/), the NAIIS was a cross-sectional, national household survey conducted in 2018 with primary objectives to estimate HIV prevalence, HIV-1 incidence, and the status of

UNAIDS 90-90-90 cascade. Complete survey methods have been described in the NAIIS final report [10]. In brief, trained survey staff collected whole blood specimens from eligible household members aged 0 to 64 years. Written informed consent of participants or caregivers was received prior to enrollment into the study.

In the household, specimens from participants aged 18 months to 64 years were tested for HIV using Nigeria's national HIV RT algorithm, consisting of Determine HIV-1/2 (Test 1 [T1]) (Abbott, California, USA) followed by Unigold HIV-1/2 (T2) (Trinity Biotech Plc., Ireland), if T1 was reactive. Statpak HIV-1/2 (T3) (Chembio Diagnostic Systems, Inc., New York, USA) was used as a tie breaker if T1 and T2 results were discordant. If T1 was non-reactive or if only T1 was reactive but both T2 and T3 were negative, the specimen was considered negative for HIV antibodies. HIV-positive diagnosis required two reactive tests. Results from the algorithm were provided to the participants in their home with appropriate counseling. At the end of each day, all specimens were transferred to a designated satellite laboratory (SL) for storage and additional testing.

Additional supplemental tests were conducted for the survey to confirm the results of specimens classified as HIV-positive by the national algorithm. At the SL, specimens determined to be HIV-positive by the national algorithm were tested by Geenius HIV-1/2 RT (Bio-Rad, California, USA). The Geenius HIV-1/2 RT assay was performed by trained laboratory technicians and the results were interpreted with an aid of an automated reader and its dedicated software provided by the manufacturer. Those with negative and indeterminate results on Geenius were further tested by HIV-1 WB (Cambridge Biotech, Maryland, USA) at the central laboratory.

To ensure the quality of household testing, the first 50 specimens tested by each household tester were re-tested at a SL using the same national algorithm. All discrepancies were adjudicated through systematic review by the data and laboratory teams. Specimen quality indicators such as hemolysis, clotted specimens, insufficient volume, and incorrect patient identification, were monitored for all specimens using the laboratory data management system (LDMS) (Frontier science, Amherst, New York).

## Ethical approval

NAIIS was approved by the Institutional Review Board of the University of Maryland, Baltimore and the U.S. Centers for Disease Control (CDC) in Atlanta, GA, USA (protocol #7103) and the Nigerian National Health Research Ethics Committee.

## Results

A total of 204,930 participants ($\geq$18 months of age) were tested by Determine HIV-1/2 RT; 199,827 (97.5%) were non-reactive and classified as negative (Fig 1, left panel). Further testing of T1-reactive specimens (N = 5,103) by T2 identified 2,886 (1.4%) as reactive and were classified as HIV-positive. Those that were T1-reactive but T2-nonreactive (N = 2,217) were further tested by T3, the tie-breaker test, yielding 72 (3.2%) HIV-positive and 2,145 (96.8%) negative. T1/T2 concordance was 56.6%, whereas 96.8% of 2,217 T2 HIV-negative specimens were confirmed as negative by T3 (Table 1).

All HIV-positive specimens (N = 2,958) by the national testing algorithm were further tested by Geenius resulting in 2,767 (93.5%) classified as HIV-1-positive, 5 HIV-2-positive (0.2%), 22 HIV-1/2 dual positive (0.7%), 83 HIV indeterminate (2.8%), and 81 HIV-negative (2.7%) (Fig 1, middle panel). HIV-2 indeterminate specimens (N = 33) were not further tested and were considered negative. Specimens that were HIV-negative or HIV-1 indeterminate by Geenius (n = 131) were further tested by HIV-1 WB (Fig 1, right panel). Of the 131 specimens,

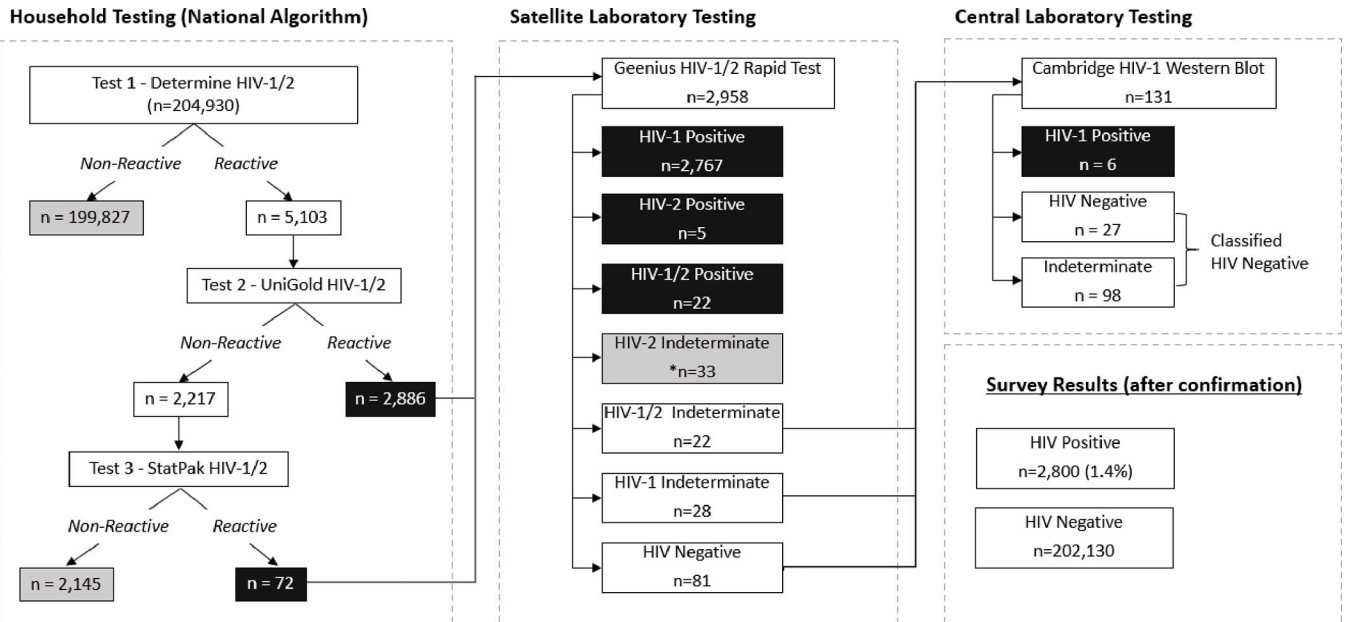

**Fig 1. Flow chart of HIV testing algorithm utilized for NAIIS 2018 for participants ≥18 months of age.** Left panel, describes Nigeria's national HIV testing algorithm, Determine HIV-1/2, Unigold HIV-1/2, and StatPak HIV-1/2 performed at the participant's household. In the middle panel, household HIV-positives were further confirmed at a satellite laboratory using the Geenius HIV-1/2 RT. The panel on the right shows results of Geenius negative or indeterminate specimens when tested by the Cambridge HIV-1 Western blot test at the central laboratory.

only 6 (4.6%) were confirmed as positive by HIV-1 WB and the remainder were either HIV-negative (n = 27) or indeterminate (n = 98). Overall, 2,800 (1.4%) specimens were confirmed as HIV-positive and 202,130 (98.6%) were HIV-negative specimens in the survey.

As a first test, the specificity of Determine HIV-1/2 in this population was 98.9% while positive predictive value (PPV) was 54.9%, when compared to the final HIV status. Overall, the PPV of the national testing algorithm was 94.5% compared to final status confirmed by Geenius and WB. The negative predictive value of the national testing algorithm was assumed to be 100%.

The protocol followed strict adherence to multiple quality indicators covering specimen collection, processing (defined as arm to freezer within 24 hours) and storage. Greater than 99.0% of specimens collected during the survey met specimen quality indicator.

## Discussion

We present here analysis of HIV testing data from more than 204,000 participants in a household survey in Nigeria. We tracked individual test results and confirmed HIV-positive diagnosis with supplemental tests (Geenius and HIV-1 WB) which allowed us to assess the PPV of the national algorithm. Our results from the NAIIS survey in Nigeria highlight some key issues

**Table 1. Performance characteristics of different HIV tests or testing algorithm, NAIIS 2018.**

| | |
|---|---|
| Test 1 and Test 2 Concordance | 56.5% |
| Test 2 and Test 3 Negative Concordance | 96.8% |
| Test 1 and Test 3 Positive Concordance | 3.2% |
| PPV (Test 1 only) * | 54.9% |
| PPV (National Algorithm) * | 94.5% |

Test 1 = Determine HIV-1/2 rapid test; Test 2 = UniGold HIV-1/2 rapid test; Test 3 = StatPak HIV-1/2 rapid test.

*Based on the final survey results.

when using national testing algorithm: 1) significantly high discordant rate between T1 and T2 of about 45% and 2) the low PPV of HIV-positive diagnosis of 94.5%. About 5.5% of individuals with positive diagnosis could not be confirmed as HIV-positive upon subsequent supplemental testing with Geenius or WB. It will be important to understand if these results from the survey can be extrapolated to HIV testing services that use the same testing algorithm in the national program.

One likely explanation for high level of discordant results between T1 and T2 and low PPV of national algorithm could be low prevalence of HIV (1.4%) in Nigeria. It is to be noted that although about 45% of Determine-reactive specimens were false-reactive, specificity of Determine was high (2,303/202,130 = 98.9%), consistent with WHO-PQ requirement of ≥98%. Use of the tie-breaker test (T3) allowed further resolution of discordant specimens in the household yielding 96.8% as negative and only 3.2% as HIV-positive (Table 1). Although most of the Determine false-reactive cases could be identified with subsequent testing, PPV of 94.5% for the national algorithm is concerning, especially in the test-and-start era. A thorough review of HTS register data from the national HIV program, along with additional prospective testing and analysis is warranted to better understand if these findings from the NAIIS can be generalized and are applicable to the national program.

Additional explanation may relate to challenges associated with trainings and monitoring of testers during the NAIIS survey. The size and scope of the survey required training of >500 individuals to conduct testing in the household which was challenging. Some of the testers were trained to perform the RT for the first time. In addition, the survey was completed in less than 6 months in a geographically diverse and big country such as Nigeria which may have affected quality of testing. Due to security reasons, we could not conduct periodic monitoring visits in the same manner as we have done in other PHIAs. Retesting a subset of specimens in the satellite laboratories for quality assurance purposes indicated a confirmation rate of about 95%, which is consistent with low PPV we observed in the survey. These considerations point to the possibility of tester-related errors leading to high discordant rates between T1 and T2 and low PPV of HIV-positive cases, keeping in mind that most false positives were eventually identified by supplemental testing.

As HIV programs mature and close the 90-90-90 gap, accurate HIV diagnosis becomes even more critical due to reduced undiagnosed HIV prevalence. Given the high coverage of ART in many LMICs and globally overall, and the declining HIV positivity, national HIV prevalence is no longer adequate to inform decisions about HIV testing strategies/algorithms and that treatment-adjusted prevalence is a more informative measure as described [11]. As recommended by WHO, the use of a 3-test strategy requiring consecutive three positive test results may be needed to achieve a PPV of 99.0% or higher when HIV prevalence is low [3]. However, addition of a third test to the national algorithm can be challenging due to additional training and quality assurance requirements, procurement, inventory management, and budget implications to the national program. In 2012, WHO recommended that all HIV-positive individuals should be retested, by another tester and using a new specimen, to verify their HIV status prior to ART initiation [12]. This can help identify tester-related error and can substantially increase PPV of HIV-positive diagnosis. Even with low PPV, the first testing event with two reactive tests is likely to result in a pool of people with HIV-prevalence of >90% among those diagnosed as HIV-positive. When this group, with HIV prevalence of >90%, is subjected to second series of tests (i.e. testing for verification), it should increase PPV to >99%, mitigating need for a third test in the algorithm. In addition, many countries are doing targeted testing including index partner testing to increase yield of HIV-positivity. These efforts have resulted in HIV-prevalence of >10% in the tested populations, thereby effectively increasing PPV of the two-test algorithm. Irrespective of what approaches are used to increase PPV, low

HIV prevalence does present a challenge in era of test-and-start. Therefore, it is incumbent upon the policy makers to ensure HIV diagnosis is as accurate as possible to reduce the likelihood of misdiagnosis and adverse impacts on the individuals and the program.

## Supporting information

**S1 Table. Raw data used for analysis.**
(XLSX)

## Author Contributions

**Conceptualization:** Hetal K. Patel, Mervi Detorio.

**Data curation:** Sunday Ikpe, Megan Bronson, Stephen Ohakanu.

**Investigation:** Ibrahim Jahun.

**Methodology:** Sehin Birhanu, Alash'le Abimiku, Mervi Detorio, Kathryn Lupoli, Daniel Yavo, Orji O. Bassey, Tapdiyel D. Jelpe, Brian Kagurusi, Nnaemeka C. Iriemenam, Divya Patel.

**Supervision:** Alash'le Abimiku, McPaul I. Okoye, Ibrahim T. Dalhatu, Sani Aliyu, Gregory Ashefor, Aliyu Gambo, Gabriel O. Ikwulono, Charles Nzelu, Isaac F. Adewole, Mahesh Swaminathan.

**Writing – review & editing:** Andrew C. Voetsch, Bharat Parekh.

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
