## [Decision Letter · Decision Letter 0]

1 Mar 2022

PGPH-D-21-01058

Performance of HIV rapid testing algorithm in Nigeria: Findings from a household-based Nigeria HIV/AIDS Indicator and Impact Survey (NAIIS)

Dear Dr. Patel,

Thank you for submitting your manuscript to PLOS Global Public Health. After careful consideration, we feel that it has merit but does not fully meet PLOS Global Public Health’s publication criteria as it currently stands. Therefore, we invite you to submit a revised version of the manuscript that addresses the points raised during the review process.

If you decide to revise and resubmit, please pay careful attention to the reviewers comments, especially reviewer 2's specific suggestions.

We look forward to receiving your revised manuscript.

Kind regards,

Alice Zwerling, PhD

Academic Editor

Journal Requirements:

1. Please amend your detailed Financial Disclosure statement. This is published with the article, therefore should be completed in full sentences and contain the exact wording you wish to be published.

State what role the funders took in the study. If the funders had no role in your study, please state: “The funders had no role in study design, data collection and analysis, decision to publish, or preparation of the manuscript.”

2. Please ensure that the funders and grant numbers match between the Financial Disclosure field and the Funding Information tab in your submission form. Note that the funders must be provided in the same order in both places as well.

3. Please update your Competing Interests statement. If you have no competing interests to declare, please state: “The authors have declared that no competing interests exist.”

4. Please provide a complete Data Availability Statement in the submission form, ensuring you include all necessary access information or a reason for why you are unable to make your data freely accessible. Note that it is not acceptable for the authors to be the sole named individuals responsible for ensuring data access.

PLOS defines a study's minimal data set as the underlying data used to reach the conclusions drawn in the manuscript and any additional data required to replicate the reported study findings in their entirety. Any potentially identifying patient information must be fully anonymized. 

If your research concerns only data provided within your submission, please write “All data are in the manuscript and/or supporting information files” as your Data Availability Statement.

5. Please provide separate figure files in .tif or .eps format only and ensure that all files are under our size limit of 20MB.

Additional Editor Comments (if provided):

Reviewers' comments:

Reviewer's Responses to Questions

**Comments to the Author**

1. Does this manuscript meet PLOS Global Public Health’s publication criteria? Is the manuscript technically sound, and do the data support the conclusions? The manuscript must describe methodologically and ethically rigorous research with conclusions that are appropriately drawn based on the data presented.

Reviewer #1: Yes

Reviewer #2: Yes

2. Has the statistical analysis been performed appropriately and rigorously?

Reviewer #1: Yes

Reviewer #2: Yes

3. Have the authors made all data underlying the findings in their manuscript fully available (please refer to the Data Availability Statement at the start of the manuscript PDF file)?

Reviewer #1: Yes

Reviewer #2: Yes

4. Is the manuscript presented in an intelligible fashion and written in standard English?

Reviewer #1: Yes

Reviewer #2: Yes

5. Review Comments to the Author

Reviewer #1: Comments for the Authors:

1. Which of the test kits was the gold standard for determining the HIV status of the subjects?

2. In this assessment why was sensitivity and negative predictive value not included?

3. What happened to the subjects that had false positive results? Was there any mechanism that recalled them for proper counseling?

4. The study had some limitations. One of them was that the national survey used some people without adequate training and experience with the test kits thus this could have contributed to the false positive cases. This should be stated. Recommendation should include to assess the performance of the test kits at health institutions where the actual testing and treatment take place.

5. The conclusions and recommendations are not clear. This should be limited to the findings in the study.

6. What are the authors recommending for HIV testing based on their findings?

Reviewer #2: This is an important paper describing the low PPV of the current 2-test HIV testing strategy employed in Nigeria, and as such it merits its publication for further dissemination. I commend the authors for sharing this important work and call for more accurate diagnosis of HIV to reduce the likelihood of misdiagnosis. I would advice authors not to be timid recommending a 3-test strategy even if retesting is in place, because this is currently the recommendation from WHO regardless of a 2-test or 3-test strategy and many countries are starting to take up this WHO recommendation. Please kindly see below some comments with the hope that this improves the current paper, it was very interesting to read this piece and I'm sure it will be of the interest of the readership of this journal.

Comments:

Lines 68–70: This paragraph needs revision because the latest WHO HIV testing guidelines in 2019 recommend that “three consecutive reactive tests should be used for an HIV-positive diagnosis” regardless of the HIV prevalence.

Line 72–74: The authors state that false positive test results in an algorithm can be reduced by using a third test or confirmatory tests including Western Blot. Considering that the latest WHO HIV testing guidelines recommend countries to discontinue the use of Western Blot given its high rate of inconclusive results, which is evidenced in the paper with a WB indeterminate rate of 74.8% (98/131), I would suggest removing WB from this sentence.

Line 75-76: authors state that Geenius is more feasible for use at the point of care. However, the interpretation of the bands by lay workers can be challenging and the use of a reader could help with this. However, the RDT reader is expensive and should be equipped with a software/laptop, which negates its use as a point-of-care, this should be mentioned.

Line 77–79: Before 2019, WHO recommended the use of a 2-test strategy in high prevalence settings and a 3-test strategy for low prevalence settings. As the study was conducted in 2018, it is important also to mention here the national HIV prevalence in Nigeria in 2018, and the rationale of using a 2-test strategy with a tie-breaker as part of the national HIV testing algorithm in a low prevalence country.

Line 97–88: please specify if Geenius HIV-1/2 RT were interpreted visually with the naked eye or with the aid of a Geenius reader with dedicated software.

Line 150: the authors suggest that one likely explanation for high level of discordant results between T1 and T2 could be the low prevalence of HIV in Nigeria (1.4%). Whilst this is one factor that could explain this finding, and it is certainly the most likely explanation for the low PPV found in this study (94.5%), authors should also highlight that discordant results between T1 and T2 are generally expected given that T1 (Determine) is optimized for sensitivity (in order to rule in all positives) while T2 (Uni-gold) has generally a high specificity (in order to rule out false positives). Given this intrinsic small difference between individual tests (T1 and T2/T3), it would be useful to understand from the authors how they define this level of discordant results between T1 and T2 to be significantly high (i.e. 45%). How does this finding compare to other published data from other settings with a similar low HIV prevalence?

Line 153–154: given all the existing evidence showing that the use of a tie-breaker, to rule in HIV infection after discordant results between T1 and T2, can cause a high proportion of false positive diagnoses (Johnson CC et al, http://dx.doi.org/10.7448/IAS.20.7.21755), authors should elaborate further why this testing strategy continues to be used in Nigeria or at least make reference to the problem of misdiagnosis when employing a tiebreaker, this is not discussed at all in the manuscript.

Line 166–168: I appreciate this observation from the authors about the possible user-related errors that could have led to a high discordant rate between T1 and T2. However, the following sentence saying that all false positives were eventually identified by supplemental testing should be corrected to “most false positives were identified” as there was still 5.5% of individuals with positive diagnosis that could not be confirmed with supplemental testing, as stated in lines 147–148.

Line 172–174: authors list several practical implementation challenges of adding a 3rd test to the national testing algorithm including additional expenses to the national programme. However, a recent modelling study shows that the incremental cost of implementing a 3-test strategy vs a 2-test strategy is negligible because the 3rd test accounts for a small and diminishing share of total HIV tests (https://www.medrxiv.org/content/10.1101/2021.03.31.21254700v1.full.pdf). Authors should mention this evidence in their paper and weigh the costs of inadvertently initiating HIV-negative persons on lifelong ART as a consequence of misdiagnosis with the use of a two-test strategy, particularly in a low HIV prevalence setting such as Nigeria.

Line 181-183: given the high coverage of ART in many LMICs and globally overall, and the declining HIV positivity, authors should highlight that national HIV prevalence is no longer adequate to inform decisions about HIV testing strategies/algorithms and that treatment-adjusted prevalence is a more informative measure as described by Tippet Bett et al https://www.ncbi.nlm.nih.gov/pmc/articles/PMC8640683/pdf/BLT.21.286388.pdf

6. PLOS authors have the option to publish the peer review history of their article (what does this mean?). If published, this will include your full peer review and any attached files.

**Do you want your identity to be public for this peer review?** For information about this choice, including consent withdrawal, please see our Privacy Policy.

Reviewer #1: No

Reviewer #2: No

---

## [Decision Letter · Decision Letter 1]

16 Jun 2022

Performance of HIV rapid testing algorithm in Nigeria: Findings from a household-based Nigeria HIV/AIDS Indicator and Impact Survey (NAIIS)

PGPH-D-21-01058R1

Dear Team Lead Patel,

We are pleased to inform you that your manuscript 'Performance of HIV rapid testing algorithm in Nigeria: Findings from a household-based Nigeria HIV/AIDS Indicator and Impact Survey (NAIIS)' has been provisionally accepted for publication in PLOS Global Public Health.

Best regards,

Alice Zwerling, PhD

Academic Editor

Reviewer Comments (if any, and for reference):

Reviewer's Responses to Questions

**Comments to the Author**

1. If the authors have adequately addressed your comments raised in a previous round of review and you feel that this manuscript is now acceptable for publication, you may indicate that here to bypass the “Comments to the Author” section, enter your conflict of interest statement in the “Confidential to Editor” section, and submit your "Accept" recommendation.

Reviewer #1: All comments have been addressed

Reviewer #2: All comments have been addressed

2. Does this manuscript meet PLOS Global Public Health’s publication criteria? Is the manuscript technically sound, and do the data support the conclusions? The manuscript must describe methodologically and ethically rigorous research with conclusions that are appropriately drawn based on the data presented.

Reviewer #1: Yes

Reviewer #2: Yes

3. Has the statistical analysis been performed appropriately and rigorously?

Reviewer #1: Yes

Reviewer #2: Yes

4. Have the authors made all data underlying the findings in their manuscript fully available (please refer to the Data Availability Statement at the start of the manuscript PDF file)?

Reviewer #1: Yes

Reviewer #2: Yes

5. Is the manuscript presented in an intelligible fashion and written in standard English?

Reviewer #1: Yes

Reviewer #2: Yes

6. Review Comments to the Author

Reviewer #1: The authors have addressed the queries raised in the first review.

Reviewer #2: Thank you to the authors for addressing all my comments and for incorporating the suggested changes. It's great to know that similar analyses are forthcoming for other PEPFAR countries, very much looking forward to read them.

7. PLOS authors have the option to publish the peer review history of their article (what does this mean?). If published, this will include your full peer review and any attached files.

**Do you want your identity to be public for this peer review?** For information about this choice, including consent withdrawal, please see our Privacy Policy.

Reviewer #1: No

Reviewer #2: **Yes: **Emmanuel Fajardo
